# Osteoblastic Cell Behavior and Gene Expression Related to Bone Metabolism on Different Titanium Surfaces

**DOI:** 10.3390/ijms24043523

**Published:** 2023-02-09

**Authors:** Eugenio Velasco-Ortega, Isabel Fos-Parra, Daniel Cabanillas-Balsera, Javier Gil, Iván Ortiz-García, Mercè Giner, Jesús Bocio-Núñez, María-José Montoya-García, Álvaro Jiménez-Guerra

**Affiliations:** 1Faculty of Dentistry, University of Seville, c/Avicena s/n, 41009 Sevilla, Spain lomonsalve Hotmail.es; 2Bioengineering Institute of Technology, Universitat Internacional de Catalunya, 08195 Sant Cugat del Vallés, Spain; 3Departamento de Citología e Histología Normal y Patológica, Universidad de Sevilla, 41009 Sevilla, Spain; 4Bone Metabolism Unit, UGC Medicina Interna, Hospital Universitario Virgen Macarena, Avda. Dr. Fedriani s/n, 41009 Sevilla, Spain; 5Departamento de Medicina, Universidad de Sevilla, Avda. Dr. Fedriani s/n, 41009 Sevilla, Spain

**Keywords:** titanium, surfaces, osteoblasts, gene expression, roughness and wettability, cell viability

## Abstract

The surface topography of titanium dental implants has a great influence on osseointegration. In this work, we try to determine the osteoblastic behavior and gene expression of cells with different titanium surfaces and relate them to the physicochemical properties of the surface. For this purpose, we have used commercial titanium discs of grade 3: as-received corresponds to machined titanium without any surface treatment (MA), chemically acid etched (AE), treated via sand blasting with Al_2_O_3_ particles (SB) and a sand-blasting treatment with acid etching (SB+AE). The surfaces have been observed using scanning electron microscopy (SEM) and the roughness, wettability and surface energy with dispersive and polar components have been characterized. Osteoblastic cultures were performed with SaOS-2 osteoblastic cells determining cell viability as well as alkaline phosphatase levels for 3 and 21 days, and osteoblastic gene expression was determined. The roughness values of the MA discs was 0.02 μm, which increases to 0.3 μm with acid attack and becomes the maximum for the sand-blasted samples, reaching values of 1.2 μm for SB and SB+AE. The hydrophilic behavior of the MA and AE samples with contact angles of 63° and 65° is superior to that of the rougher samples, being 75° for SB and 82° for SB+AE. In all cases, they show good hydrophilicity. GB and GB+AE surfaces present a higher polar component in the surface energy values, 11.96 and 13.18 mJ/m^2^, respectively, than AE and MA, 6.64 and 9.79 mJ/m^2^, respectively. The osteoblastic cell viability values at three days do not show statistically significant differences between the four surfaces. However, the viability of the SB and SB+AE surfaces at 21 days is much higher than that of the AE and MA samples. From the alkaline phosphatase studies, higher values were observed for those treated with sand blasting with and without acid etching compared to the other two surfaces, indicating a greater activity in osteoblastic differentiation. In all cases except in the Osterix (Ostx) —osteoblast-specific transcription factor—a decrease in gene expression is observed in relation to the MA samples (control). The most important increase was observed for the SB+AE condition. A decrease in the gene expression of Osteoprotegerine (OPG), Runt-related transcription factor 2 (Runx2), Receptor Activator of NF-κB Ligand (RANKL) and Alkaline Phosphatase (Alp) genes was observed in the AE surface.

## 1. Introduction

It has been studied by several authors that the topography of titanium surfaces and their alloys affect bone formation, protein adsorption and osteoblast interaction with the biomaterial. The expression of a specific pattern of integrin receptors on the osteoblast membrane has been observed [1,2]. Bone formation is initiated by type 1 collagen generated from osteoblasts, being the most abundant protein in the extracellular matrix. Collagen 1 provides a scaffold for the deposition of the mineral component of bone apatite, which gives it mechanical strength. The organic structure provides elasticity [3]. Bone homeostasis is governed by several signaling pathways that will trigger cell differentiation [4]. These strategies are due to fibroblast cell growth factors (FGF), bone morphogenetic protein (BMP) and Wnt signaling expression [4,5]. Osteoblast cells express two transcription elements: Runt-related transcription factor 2 (RUNX2) and osteoblast-specific transcription factor (Osterix or OSX). These factors are critical and indispensable for osteoblast differentiation [6,7]. Some authors determined that RUNX2 and OSX have a fundamental role in osteogenesis but not in cell differentiation [8,9].

Osteogenesis and angiogenesis are mechanisms that are related to bone growth, remodeling and repair [7,9,10,11]. Osteoclast cells activate angiogenesis through the expression of proangiogenic factors, including vascular endothelial growth factor-A (VEGF-A) [11,12,13]. This factor functions with receptor activator of nuclear factor kappa-B receptor ligand (RANKL) to promote osteoclastogenesis [11]. Deckers et al. confirmed that angiogenesis is caused by osteoblast quiescence and not so much by osteoclast cell activity [12]. In addition, Cackowski et al. demonstrated that osteoclast activity increased angiogenesis and osteoclast cell inactivity elicited osteoprotegerin (OPG), decreasing angiogenesis. Increasing osteoclastic activity with parathormone (PTH) produced an increase in angiogenesis [13].

It has been possible to determine the influence on osteoblast adhesion, proliferation and differentiation with different implant topographies in both in vitro and in vivo studies [14,15,16]. One of the most significant results was how commercially pure and roughened titanium favored the expression of alkaline phosphatase (ALP) and collagen type in the same way titanium with large crystalline grain size behaved [17]. The roughness’ influence on osteoblastic cell behavior, especially differentiation, is caused by the action of protein kinase A and PL A2 [18], and by integrin-generated signaling [15]. Other aspects favoring differentiation on rough surfaces were the subsequent expression of cytokines and growth factors mediated by osteoblastic cells. It could be seen in in vitro studies performed on rough titanium with osteosarcoma cells (MG-63) that an increase in TGF-β and IL-1β occurred [15,19,20,21,22]. A decrease in proliferation was observed on titanium surfaces with roughness greater than 2.5 μm Ra due to a prostaglandin-mediated response. An increase in cellular phenotypic markers of differentiation (ALP activity, osteocalcin) could also be observed.

These aspects play a fundamental role in the generation of new osteoblastic cells and in the osteoconduction mechanism. In addition, titanium roughness leads to an increase in the osteoblastic differentiation process through variations in transcriptional regulation or gene expression of key osteogenic factors due to morphological variations of the cells caused by the topography of the titanium surface. It is now important to determine the influence of titanium topography on bone formation. There are many variables involved and the topography and physicochemical properties of the surface are key. At present, there is a serious problem with peri-implantitis: there is an important loss of bone that in many cases causes the loss of the dental implant. It is important to be able to obtain a titanium surface which eliminates bacteria growing and stimulates bone regeneration.

The aim of this work is to study the osteoblastic behavior and its gene expression in different topographies of titanium dental implants. The aim and originality of this work is to determine the influence of roughness and its surface properties on bone metabolism. In this way, it will be possible to discriminate the contribution of the different variables since, at present, only studies of rough surfaces have been carried out via shot blasting with alumina particles without combining them with other treatments, such as etching.

## 2. Results

The topographies of the different surfaces studied can be seen in Figure 1. It can be observed that the machining discs (MA) have the reliefs of the machining but the typical valleys and pits of the discs treated with abrasive particle projection (SB) and abrasive particle projection with acid etching(SB+AE) are not appreciated. In the latter case, it is observed that the acid attack of the mixed treatment produces a very small variation in the topography. The most important contribution of the macroroughness is due to sand blasting and the acid attack produces a more microstructural attack. This fact can be verified by observing the topography of the discs in which only the acid attack (AE) has occurred. 

The roughness of the different surfaces was determined via interferometric microscopy in order to obtain the different parameters that characterize the roughness, as can be seen in Table 1. The MA discs show the lowest values of Ra. This roughness is defined as the average pico–valley roughness and the values can be considered a smooth surface. The discs that have been treated with sand-blasting SB and SB+AE have the highest Ra values and there are no statistically significant differences between them. These results show the small contribution to roughness of acid attack. The AE discs show a roughness higher than MA and lower than that corresponding to the SB and SB+AE discs. The roughness of SB and SB+AE discs presented statistically significant differences compared with the AE discs (*p* < 0.005).

The topographies observed at higher magnifications using FESEM showed that the discs treated with MA had a wavy surface with grooves due to the machining machine. The surfaces of the other discs were rougher and more irregular, especially the samples that had been sand blast treated. In particular, the SB discs revealed large peaks and valleys of varying geometry with numerous planes, while the surfaces of the other Ti discs were less sharp. The discs exposed to SB+AE procedures showed the highest roughness and a heterogeneous micropitting surface. The discs subjected to AE exhibited micropitting of different sizes. The surfaces observed in the electron microscopy images matched those of the 3D representations obtained from optical interferometry in the lower panel of Figure 1.

The water contact angles (CA), and the calculated values for the surface free energy (SFE) and its compounds following the Owens and Wendt approach are shown in Table 2 and Table 3. Overall, the grit-blasting treatment decreased surface wettability, i.e., increased CA. This effect was particularly pronounced for those surfaces grit-blasted with residual particles of Al_2_O_3_. 

When comparing dispersive or polar components of SFE, there was a general trend in polar components to decrease when the samples contained alumina (Table 3) [19]. Statistically significant differences in the polar components of rough surfaces with alumina with respect to control and clean surfaces were determined for the samples treated with the largest particles. 

Figure 2 shows the cell viability for two time points, 3 and 21 days, using osteoblastic cells. The viability measure has been obtained by determining that its reduction is proportional to the increased osteoblastic cell viability of the different surface types. The control has been taken as the MA disc since it is the as-received one, showing how all three surfaces improve viability with respect to MA. A similar viability can be observed for the three days in the titanium discs with surface treated with alumina shot blasting and acid etching. No statistically significant differences were observed between SB, SB+AE and AE cell viability at 3 days with a *p* < 0.001. Differences were observed with MA of the other surfaces with a *p* < 0.001. At 21 days, we have the best viability value for the surfaces that have only been treated with shot blasting. Figure 3 shows the osteoblastic cells on the different surfaces after 3 and 21 days. 

In Figure 4, the osteoblasts on the MA surface (a) and on the surface (SB+AE) can be observed at higher magnifications. It can be observed that the morphology of the cells is flat, with greater activity on the rough SB+AE surface where there is greater dorsal activity with filopodia and some whitish nuclei on the surface, which could be attributed to the permineralization activity of the osteoblastic cell. 

Figure 5 shows the results of alkaline phosphatase, indicating the differentiation of osteoblastic cells experiencing the increase in this enzyme. It can be seen that on day 3, the cells are not fully differentiated and the values of pNPP are lower than on day 21. The surfaces with the greatest differentiation are the discs that have been subjected to shot blasting with or without acid etching treatment. 

Figure 6 shows the gene expression with respect to the M condition, which we consider to be the control condition. It can be seen that in all cases except in the Ostx, a decrease in gene expression is observed. The increase was observed for the SB+AE condition. A decrease in gene expression of OPG, Runx2, RANKL and Alpl genes was observed in the acid attack discs. There is a higher expression with respect to the Ostx control in the discs treated with shot blasting and acid etching. For the RunX2 gene, there is an increase in expression on day 21 with respect to day 4.

## 3. Discussion

Currently, dental implants with a machined or polished surface are not used because roughness favors osteoblastic activity, increasing the degree of osseointegration. Roughness causes a very important increase (between 5 and 8 times that of the real machined surface), increasing the bone–titanium junction and consequently increasing the mechanical fixation. Some so-called hybrid implants have been developed, which have the polished part in the area of the connection and the first two or three coils of the dental implant. This design was created to try to avoid the formation of biofilm in peri-implantitis processes since the bacterial colonization is lower in the polished parts of the dental implant [16,17,18]. 

The acid attack is a methodology that slightly increases the roughness of the dental implant. However, as we have been able to verify in the results, the increase is small. It would be a mistake to treat it with acids of higher concentration since the roughness will increase vertically but not horizontally (distance between peaks) and this increase in depth does not favor osseointegration. In addition, the increase in concentration could produce an incorporation of hydrogen in the titanium, causing the formation of hydrides and causing hydrogen embrittlement, drastically reducing the mechanical properties. In this case, the implant should be subjected to a hydrogen elimination heat treatment consisting of heating at 150 °C for 1 h [20,21].

The sand-blasting treatment with alumina particles is the most common treatment for dental implants as it allows the optimization of the roughness of the dental implant to the affinity of the osteoblastic cells. The size of the abrasive particles, the nature of these particles, the impact pressure, and the distance between the gun and the surface can be modulated to obtain the optimal roughness. As we have seen, the subsequent acid attack does not significantly increase the roughness, but it does create a microroughness in the macroroughness obtained by sand blasting, which will improve the biological behavior. Increasing the particle size, the abrasiveness of the particles, the impact pressure or decreasing the distance between the gun and the surface would increase the roughness. In any case, higher roughness would facilitate bacterial colonization. This is the reason why the optimum roughness (Ra) values should range from 0.9 to 1.9 μm [16,18].

These topographic changes have their influence on the contact angles; as we have been able to observe, the granulated materials increase the contact angle and the polar component of the surface energy, aspects that favor the biological behavior. In addition, sand-blasting treatments cause a compressive tensional state on the surface that significantly improves the resistance of the dental implant to fatigue, as has been demonstrated by various authors. This state of surface compression, which causes a delay in the nucleation of a crack on the surface due to cyclic masticatory loads, is not present in machined or acid-etched implants. For this reason, these implants will have a worse mechanical behavior in the long term [20]. 

The discs do not present porosities since the discs are made of cold-worked titanium, which is the same material used to manufacture dental implants. Therefore, it is only the roughness that influences the biological behavior and it has been observed that the roughness established via sand blasting and acid attack is the one that gives a higher viability of the osteoblastic cells. This roughness has been observed to vary the degrees of hydrophilicity and the polar and dispersive components of the surface energy that favors osteoblastic viability and bone gene expression. The effect of alumina debris on the surface has been demonstrated in several studies [20,21,22,23]. In a study [22] with the same type of discs and surfaces, it was observed that the X-ray energy dispersive microanalyses showed the presence of aluminum and oxygen in the SB samples due to the use of Al_2_O_3_ as abrasive particle for the sand-blasting treatment. These particles are embedded on the surface of the dental implant. This also occurs in the SB+AE samples, although the amount of alumina decreases slightly due to the partial dissolution of the particles. 

It is well-known how the roughness of titanium affects wettability, protein adsorption on the surface and surface energy in both the polar and dispersive components, as well as the zeta potential. These physicochemical properties are key factors for cell migration, proliferation and differentiation [22]. The mechanism of osseointegration is based on the interactions of the osteoblastic cells with the surface of the biomaterial and the generation of collagen scaffolds in which the apatite and other organic components will be deposited for the formation of the so-called bone matrix and, consequently, the formation of new bone tissue. We can assure that the adhesion of osteoblasts and their cellular activity together with the formation of the extracellular matrix of cells (ECM) are key for the subsequent stages of proliferation and differentiation to complete the osseointegration process [24,25,26,27,28]. 

It should be noted that the roughness of SB+AE favors osteoblastic cell behavior but also favors bacterial colonization [19,20,22]. Roughness favors biofilm formation and a compromise roughness must be reached for good cell viability and the worst for bacterial adhesion.

It can be said that osseointegration presents four distinct stages. The first corresponds to the wettability of blood with titanium. In this case, depending on the topography and physicochemical characteristics, it will have a more hydrophilic or hydrophobic character. It is interesting to note that the contact angle is not stable over time because the proteins in the blood solution adsorb on the titanium surface. This is the second stage of osseointegration, which corresponds to the nonspecific adsorption of proteins. The adsorption of proteins on titanium causes the liquid nature with respect to the solid to change as the proteins from the blood are adsorbed. It can be seen how the contact angle increases and the hydrophobicity consequently increases with time. This second stage is critical since it will depend on the proteins that are adsorbed and will provoke the bacterial migration of osteoblasts or fibroblasts depending on the type of proteins that have been adsorbed. This fact means that this second stage of non-specific adsorption of proteins tries to be changed by the specific adsorption of proteins and that the specificity is in precursor proteins of the osteoblastic cells. The third stage will be cell migration with the three proper and sequential activities: adhesion, proliferation and differentiation; the fourth is the formation of bone tissue that will cause metal–implant fixation and, consequently, osseointegration. Therefore, adsorption of adhesive matrix proteins from the surrounding medium, followed by the recognition of these proteins by cells [29,30], triggers specific cellular responses [31,32]. 

It has been studied how biomaterial properties, such as wettability [33], surface electrical charge [34], chemistry and surface topography [35], play a key role in the establishment of cell–biomaterial contacts [36]. Although there are controversial results [37], the fact that the surface topography strongly influences the behavior of adherent cells is widely accepted in the literature [31,32,33,34,35], and, in particular, for titanium surfaces [29,37]. It has been possible to observe how fibroblasts adhere on titanium following the machining grooves in an aligned manner and avoiding chaotic adhesion on the metal surface. This fact has made it possible to make dental implant collars in which the grooves are placed in circles, and it has been possible to observe the annular growth of the soft tissue imitating the natural configuration that occurs in natural teeth. This sensitivity of fibroblasts to these topographies allows for guided cell growths that, in this case, act as a biological seal to prevent bacterial infiltration [38]. Osteoblasts also have a sensitivity with machining groove topographies but with much less intensity, and guided growth is more difficult in the case of fibroblasts [39,40,41,42].

Unfortunately, the mechanisms explaining the response of osteoblasts with topography are still not very clear and the important role played not only by roughness values but also by surface energy values, especially polar contribution, can be confirmed. It has been confirmed that concave valleys of a Ti surface accumulate a higher density of negatively charged hydroxyls and highly polar groups are located at convex peaks [43,44]. The possibility of air pockets at the bottom of the topography in the early stages of contact with the protein solution, due to dynamic effects during wetting, is a possible explanation for the preferential accumulation of protein at the top of the topographic features. 

It should be noted that in sand-blasting treatments with abrasive particles, some are included in the surface. These abrasive particles—usually alumina—influence the wettability and surface energies in both the dispersive and polar components and probably influence the distribution of adsorbed fibronectin on the SB and SB+AE surfaces. In addition, these alumina particles may have affected the total amount of adsorbed protein through their influence on wettability and SFE [41,42]. Similar adsorption patterns were observed in some papers [43,44] for other proteins, such as albumin and fibrinogen, i.e., one globular and one fibrillar protein. They did not observe any correlation between jet particle size, i.e., surface roughness, and corresponding changes in ELA [43,44,45]. All these factors lead to the conclusion that the observed heterogeneity in protein adsorption must be attributed to a property or characteristic of the Ti sand-blasted surface, independent of the protein structure or the way the roughness is formed [17].

Indeed, our results agree with the studies of [19,21,46,47], which showed that cells cultured on rougher surfaces tended to show more differentiated osteoblast attributes than cells cultured on smoother surfaces. ALP activity was much higher (*p*-value < 0.001) on SB+AE and SB surfaces than on all M or AE throughout the experimental period (Figure 3). Significant differences were observed among all samples reaching the maximum ALP activity between 3 and 21 days of culture. 

According to the results of Zhao et al. [48], SB and SB+AE surfaces showed higher viability of osteoblastic cells compared to machine-made and acid-etched surfaces. These authors established that surfaces with high surface energy (especially the polar component) may favor the selection of cells at a more advanced stage of differentiation; his fact is confirmed by our research. On the other hand, Lai et al. [49] demonstrated that the higher surface energy of titanium surfaces improved cell metabolism in the initial phase of the cell response and could act by influencing the expression of adhesion-associated molecules. However, no significant differences in the gene expression of CK14, integrin 6, integrin 4, vinculin, TGF-1 or TGF-3 were observed between epithelial cells (HSC-2) cultured on MA, SB and SB+AE [27]. Hallab et al. [50] demonstrated that ELA is a more relevant surface characteristic than surface roughness for cell adhesion strength and proliferation, and that the surface energy components of the different materials tested were shown to be related to cell adhesion strength: a poor correlation was observed between the dispersive component of ELA and adhesion strength when compared to the polar component of ELA. Similar correlations were observed by Martelet and Ponsonet et al. [33,51].

The higher alkaline phosphatase levels of the SB and SB+AE samples indicate increased cellular activity compared to the MA and AE surfaces, a finding that suggests the presence of mature secretory osteoblasts. According to Zhao et al. [48], high surface energy increased osteocalcin production. These results are in agreement with those of Xavier et al. [52], who showed that calcified nodule formation was significantly reduced on machined and acid-etched surfaces. One explanation for this reduction could be the release of titanium constituent ions, which interfere negatively in the mineralization process. 

From the results, it can be observed that roughness is the main factor influencing osteoblastic activity, as well as its genetic expression. Acid-etching treatment does not provide the dental implant surface with substantial improvements in osteoblastic activity. Shot-blasting treatment has been shown to increase the contact angle and decrease the surface energy. The roughness values of shot-blasting treatments favor osteoblast adhesion and subsequent proliferation and differentiation. Likewise, alumina residues favor osteoblastic activity, as described by Gil et al. [22].

The present work has a series of limitations and perhaps it would be appropriate to carry out a study at shorter times of adhesion and proliferation of osteoblastic cells to better understand the behavior at short times [26,53]. In any case, it is difficult to discriminate the influence of all the variables on the cellular and genetic behavior of osteoblastic cells: chemical nature, topography, residual stress, surface energy, zeta potential, and wettability, among the most important ones. Györgyey et al. [53] showed that the large variations that the properties of titanium can affect biological behavior. This work is intended to help understand the influence of surfaces on osteoblastic behavior, but further research is needed to understand the biology of bone formation and its integration with implants.

## 4. Materials and Methods

Commercially pure grade 3 titanium discs were studied. The dimensions of the discs were a thickness of 4 mm and a diameter of 12 mm. The materials were donated by the Galimplant Dental Implant Company (Sarria, Spain). The surface treatments performed are those that correspond to those performed for commercial dental implants. They were classified into four groups according to surface treatment.

Machined (MA). The discs show the mechanical abrasion typical of the machining of dental implants without subsequent surface treatment.

Sand blasted and acid etching (SB+AE). The discs were blasted with alumina particles of 250 to 450 μm at a pressure of 2.5 bar and at a gun–surface distance of 100 mm. They were then washed with distilled water and treated via immersion in a 1:1 acid mixture of concentrated HCl and HNO_3_ for 45 s. 

Sand blasted (SB). Only the exposed sand-blasting process was performed on the SB+AE. In this batch, the acid etching was not performed, only the sand blasted with alumina. It could be determined that about 3% of the abrasive alumina particles were included in the rough titanium surface, as is usual in dental implants sand blasted with alumina.

Acid etching (AE). Acid etching was performed with a mixture of 1:1 concentrated HCl and HNO_3_ acids for 45 s. 

The cleaning of the samples for the surface characterization studies: roughness and machinability was carried out with methyl alcohol for 15 min in ultrasound and then in acetone for 5 min. Drying is carried out with hot air flow.

The topography of the discs was observed using scanning electron microscopy (SEM) with a Zeiss Neon40 FE SEM (Carl Zeiss NTS GmbH, Jena, Germany). Sputtering was not necessary to enhance the electrical conductivity. An electronic acceleration potential of 5KV was used. The working distance was 7 mm. 

Roughness was determined using ten samples for each surface studied. Three places for each sample were analyzed (4 surfaces × 10 samples/surface × 3 places/sample). A Wyko NT1100 interferometric microscope (Veeco Instruments, Plainview, New York, USA) using white light and beam orthogonal to the surface was used. Constant magnification was used with a 5× magnification objective. Roughness was determined on a scan area of 1 × 0.5 mm and 10 values were obtained for each surface in different areas. The arithmetic mean height (Ra), surface asymmetry (Rsk), which measures the asymmetry of the profile height distribution, and surface kurtosis (Rku), which represents the sharpness of the height distribution or “peakedness” of the profile [23,24], were studied. The results were analyzed using Wyko Vision 232TM software 3Xb (Veeco Instruments Plainview, New York, USA). 

The wettability of the four surface types was determined using the static contact angle technique following the sessile drop technique under static conditions and at a temperature of 25 °C [25]. Six samples for each treatment were analyzed. 

These measurements were carried out with an OCA15 Plus Instrument (Dataphysics, Filderstadt, Germany). The drop volume was 3 µL and the dosing rate was 1 µL/min. The fluids used were ultrapure distilled water (Millipore Milli-Q, Merck Millipore Corporation, Burlington, MA, USA) and diiodomethane (Sigma-Aldrich, Barcelona, Spain). 

Determination of the surface free energy was calculated with the Owens and Wendt equation [26].
(1)γL1+cosθ=2γLdγSd12+γLpγSp12 
where *γ^d^* and *γ^p^* are dispersive and polar components, respectively. *γ_L_* corresponds to the of the liquid surface tension and *θ* is the contact angle between the liquid (*L*) and the solid (*S*).

Data was analyzed with SCA 20 software 3240 (Dataphysics Filderstadt, Germany,). Three measurements were carried out for three different samples in each series.

Osteoblastic cells (SaOS-2; ATCC, Manassas, VA, USA) were used for in vitro studies. They were cultured in Dulbecco’s modified Eagle medium (DMEM) and McCoy’s modified 5A medium. To these two solutions, we added 10% fetal bovine serum (FBS) and 50 µg/mL l-glutamine and penicillin/streptomycin at a concentration of 2 mM (Invitrogen, Carlsbad, CA, USA). Cultures were grown at 37 °C (body temperature) in a 5% CO_2_ incubator under humidified conditions. 

Confluent cells were separated from the culture via incubation with TrypLE (Invitrogen, Carlsbad, CA, USA) for 1 min. The cell dilution was centrifuged and placed in a new culture medium. Subsequently, the titanium discs were seeded on the different surfaces with a cell concentration of 5000 cells per each of the discs studied/disc and incubated at 37 °C. After 3 and 21 days of incubation, the cells were lysed with 200 µL/well of M-PER^®^ (Pierce, Rockford, IL, USA). Cell proliferation was determined on the different topographies of the titanium discs using the LDH cytotoxicity detection kit (Roche Applied Science, Mannheim, Switzerland). The number of proliferating cells is proportional to the release of lactate dehydrogenase (LDH) and this lactate was determined spectrophotometrically at 492 nm with a conventional ELx800 microplate detector (Bio-Tek Instruments, Inc., Winooski, VT, USA).

Alkaline phosphatase (ALP) activity of osteoblastic cells was determined using Sensolyte pNPP alkaline phosphatase colorimetric assay (Anaspec, Fremont, CA, USA). For ALP determination, a wavelength of 405 nm was used and detection was performed with a conventional ELx800 microplate reader (Bio-Tek Instruments, Inc., Winooski, VT, USA). 

The evaluation of osteoblastic gene expression was performed with the determination of total RNA at different culture times, which was extracted using the RNeasy^®^ Mini Kit (Qiagen, Hilden, Germany). RNA was quantified using the NanoDrop ND-1000 spectrophotometer (NanoDrop Technologies Montchanin, DE, USA). One hundred nanograms were retrotranscribed to cDNA using the QuantiTect reverse transcription kit (Qiagen). The cDNA products were diluted to 1 ng/µL and used as real-time quantitative polymerase chain reaction (RT-qPCR) templates. Primers were selected from the Universal ProbeLibrary (Roche Applied Science) to amplify osteoblast differentiation specific genes. Primers for genes presenting more than one transcript were selected from common regions. SYBR Green RT-qPCR analyses were performed using the QuantiTect SYBR Green RT-PCR kit (Qiagen, Hilden, Germany) on a StepOnePlus real-time PCR machine (Thermo Fisher Scientific, Waltham, MA, USA). The specificity of each RT-qPCR reaction was determined using melting curve analysis.

## 5. Conclusions

The viability of osteoblastic cells and alkaline phosphatase levels indicating cell differentiation are higher on titanium surfaces that have been sandblasted with and without acid etching. This improved cell behavior is due in part to the higher polar component of the free surface energy. In all cases except in the Ostx, a decrease in gene expression is observed in relation to the control samples. The most important increase was observed for the sand blasted with acid etching surface. A decrease in the gene expression of OPG, Runx2, RANKL and Alpl genes was observed in the acid etching surface. These results suggest that the treatments used in the present study may support favorable biological responses in vivo.

## Figures and Tables

**Figure 1 ijms-24-03523-f001:**
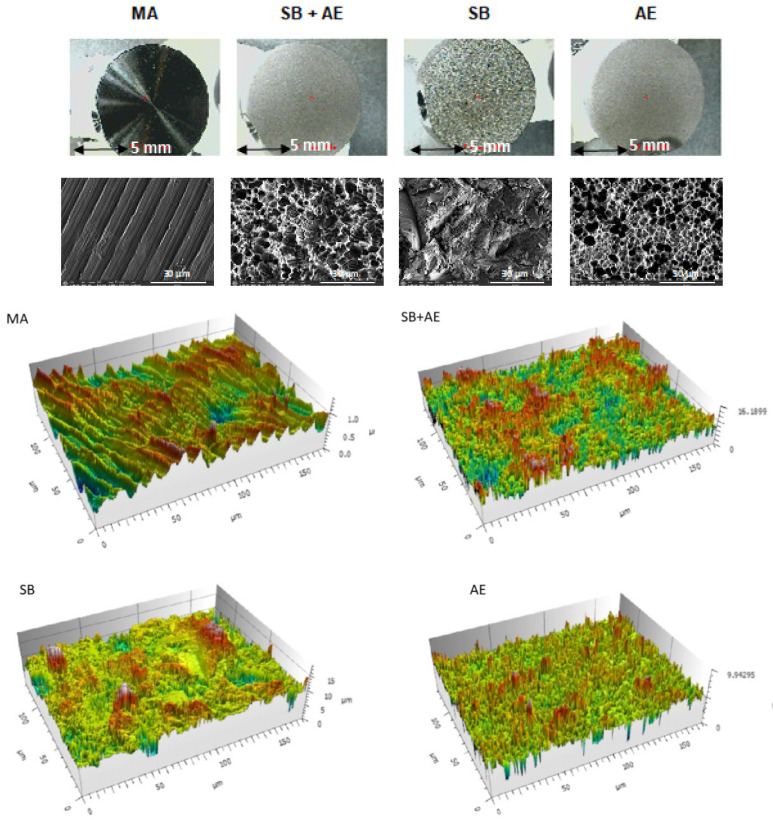
Macroscopic surfaces, topography observed using a scanning electron microscope and perfilometry of the different surfaces obtained (MA: machined; SB+AE: sand blasted with acid etching; SB: sand blasted; and AE: acid etched samples).

**Figure 2 ijms-24-03523-f002:**
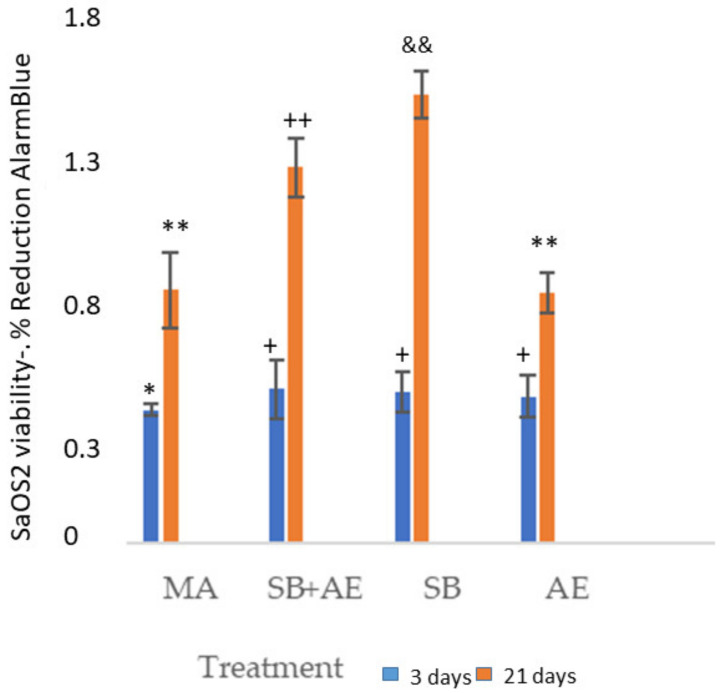
SaOS-2 osteoblastic cells viability at 3 and 21 days for the different topographies studied. Each symbol indicates the statistical differences’ significance *p* < 0.001.

**Figure 3 ijms-24-03523-f003:**
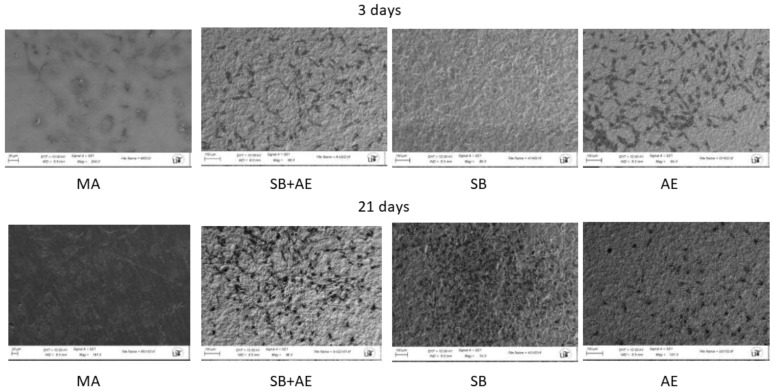
SaOS-2 osteoblastic cells observed by SEM at 3 and 21 days for the different topographies studied.

**Figure 4 ijms-24-03523-f004:**
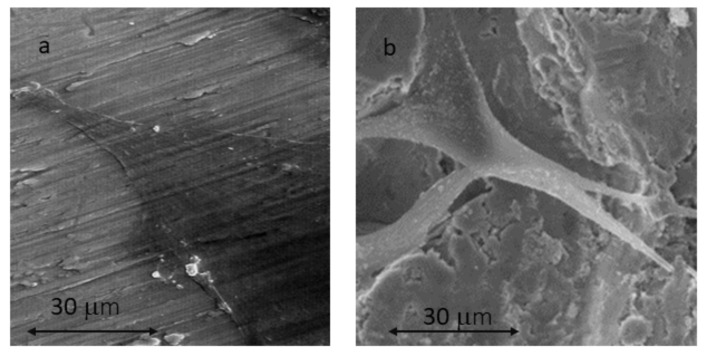
SaOS-2 osteoblastic cells observed using SEM at 21 days for MA surface (**a**) and SB+AE surface (**b**).

**Figure 5 ijms-24-03523-f005:**
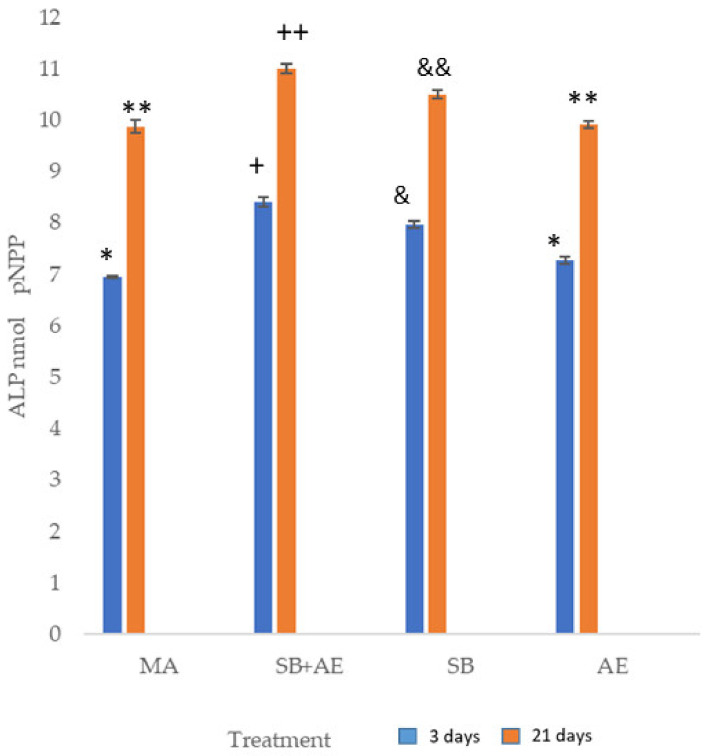
Alkaline phosphatase of SaOS-2 osteoblastic cells at 3 and 21 days for the different topographies studied. Each symbol indicates the statistical differences’ significance *p* < 0.001.

**Figure 6 ijms-24-03523-f006:**
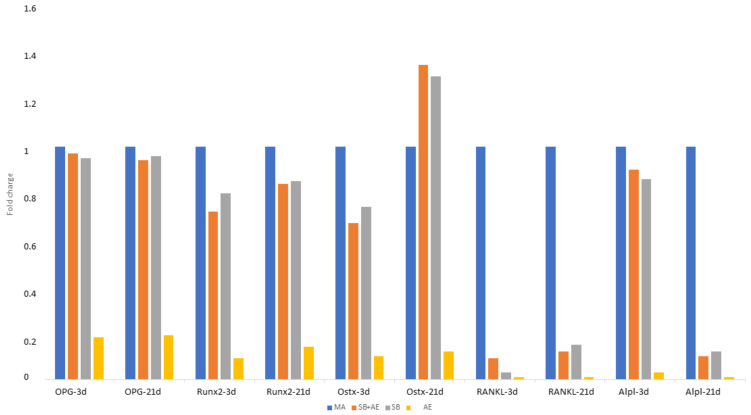
Gene expression of SaOS-2 osteoblastic cells at 3 and 21 days for the different topographies studied.

**Table 1 ijms-24-03523-t001:** Mean ± standard error of the mean of the surface roughness parameter Ra for the different types of Ti implants. (* and ** means statistically differences significance).

Ti-Disc	MA	SB+AE	SB	AE
Ra (µm)	0.026 ± 0.008	1.235 ± 0.020 *	1.162 ± 0.492 *	0.303 ± 0.112 **

**Table 2 ijms-24-03523-t002:** Apparent contact angles for the three liquids used on the different c.p. Ti surfaces. Values are mean ± standard error of the mean. Statistical differences vs. smooth surfaces for each column are indicated by single and double asterisk symbols (*p* < 0.05).

Surface	Water CA’ [°]	Di-Iodomethane CA’ [°]	Formamide CA’ [°]
MA	61.9 ± 5.0	48.0 ± 2.9 *	51.0 ± 1.6
SB+AE	81.9 ± 5.1 *	36.2 ± 3.0 **	36.0 ± 1.3 *
SB	76.7 ± 6.5 *	56.9 ± 1.7	58.9 ± 2.0
AE	63.3 ± 8.1	37.6 ± 4.0 **	33.9 ± 5.0 *

**Table 3 ijms-24-03523-t003:** Water contact angle, surface free energy and its components for the different Ti surfaces. Values are mean ± standard error of the mean. Statistical differences vs. smooth surfaces for each column are indicated by single and double asterisk symbols (*p* < 0.05).

Surface	Surface Free Energy (mJ/m^2^)		
	Total Surface Free Energy	Dispersive Component	Polar Component
MA	42.98 ± 1.70	33.19 ± 1.94 *	9.79 ± 2.93 *
SB+AE	42.48 ± 1.88	29.30 ± 1.22 *	13.18 ± 1.20 **
SB	42.95 ± 1.69	30.99 ± 0.85 *	11.96 ± 0.90 **
AE	47.08 ± 2.92 *	41.10 ± 2.34 *	6.64 ± 3.15 *

## Data Availability

The authors can provide details of the research upon reasonable request.

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
