# Peer review of "Osteoblastic Cell Behavior and Gene Expression Related to Bone Metabolism on Different Titanium Surfaces"

_ijms, 2023, doi:10.3390/ijms24043523_

Round 1

Author Response

REVIEWER 1.

Dear Reviewer,

Thanks for taking the time to review our manuscript and suggest to us to improve our work by providing a lot more detail. We have done so, and we are now submitting a manuscript that not only addresses the points you specifically raised but also many others that we have considered in order to deliver what we think is a much improved version of our work. This version includes more paragraphs, English grammar revisions in all main sections, new references. Thanks a lot and happy new year. We are looking forward to your comments.

Sincerely,

Francisco-Javier Gil Mur

Dear Authors, please correct the following parts:

1- Introduction, page 2: what do you mean by rough titin? Did you mean Titanium?

Yes. It is a mistake. The mean is titanium, the word has been changed.

- Materials and Methods:

2- Why did you test grade 3 Ti, as nowadays CP grade 4 or Ti6Al4V alloy is used for

manufacturing dental implants?

Currently, 90% of dental implants are made of grade 3 titanium, although sometimes the grades between 3 and 4 are very similar due to the interstitial content. We can also confirm that the differences in grade between 3 and 4 have no influence on the biological behavior and only show differences in the mechanical behavior, where the increase of interstitial elements increases the hardness and mechanical resistance. Dental implants manufactured with Ti-6Al-4V alloy are a minority, being approximately 7% of the dental implants and 3% those manufactured with other materials such as zirconia or PEEK.

3- The applied surface treatments are the same as used in case of dental implants? If

yes, please add it to the manuscript.

Done

4- SB samples description…not clear, the samples had acid attack?

This treatment has been explained according to the reviewer comment.

5- It would have been good to compare the samples to a positive control, for e.g. a

plate (see Figure 3. in Masa R. et al: J. Funct. Biomater. 2022, 13, 202).

The MA samples have been used as the titanium control since they are the untreated surfaces and the ones referred to in the scientific literature for these studies. We have added the reference of R, Massa et al. (25) as it is an excellent work where readers can expand on the details of these techniques in titanium wettability testing,

6- Please give the cleaning procedure of the samples which was used before the SEM,

roughness measurements, CA and in vitro analysis.

Done. A new paragraph has been introduced.

7- Dynamic CA measurements were performed or static ones?

Static, this aspect has been introduced in the text.

8- In my opinion 24 h cell attachment is also important to measure.

The reviewer is right that 24 hours would have been interesting. However, the authors followed the references of other authors and followed the 3 days and 21 days since 24-hour values are considered low. The 3-day and 21-day studies show the progression although this is undoubtedly a limitation of the work that has been introduced in the discussion.

- Results:

9- Every data is given as mean ± SD. SD is not correct in these cases, as standard error

of the mean (SEM) is relevant for describing the error of the mean values. Please give

SEM values instead of SD!

The standard deviations have been deleted and in the actual version the values are standard error of the mean according to the reviewer.

10- Please correct the subdivision of the values on Y axis and write dot instead of

comma.

Done

11- Please highlight the significances between the different means in Figures 2. and 4.

with bars and asterisks.

Done

12- Why did you measure gene expression at day 4 and not 3 as for the cells?

The gene expression was measured at day 3 as for the cells. The mistake has been corrected.

13- If you have several factors (type of treatment, surface roughness and CA) which

influences the proliferation of the SaOS-2 cells, it would be advisable to put them on

the same figure (see for e.g. Figure 4. in Györgyey Á et al: J Biomater Appl. 31(1):55-

67, 2016).

In agreement with the reviewer, a paragraph has been added in the discussion of results explaining the complexity of studies of the biology of osteoblast adhesion, proliferation and differentiation with so many variables that influence osteoblast adhesion. This explanation is accompanied by the bibliographic citation suggested by the reviewer.

Reviewer 2 Report

The article is based on the study of the effect of titanium surface modifications on osteoblasts, ALP alkaline phosphatase activity as well as osteoblastic gene expression.

Overall, the work is interesting and quite well-written. It lacks information on the novelty of the research described therein. Moreover, it seems to me that the work should show not only single samples of titanium machined, sand-blasted and acid etching, but also the influence of the conditions of these modifications on the surface of titanium.

The results are well described, and the discussion is interesting.

There should be a Y-axis in the charts so that you can better track the values.

The authors need to improve the way of representing decimal numbers - in English, decimals are written with a period, not a comma.

Authors should also standardize the bibliography - sometimes the names of journals are given in abbreviation, sometimes as full names.

In general, I believe that authors wishing to publish a paper in the International Journal of Molecular Sciences should take into account the comments presented in this review.

Author Response

REVIEWER 2

Dear Reviewer,

Thanks for taking the time to review our manuscript and suggest to us to improve our work by providing a lot more detail. We have done so, and we are now submitting a manuscript that not only addresses the points you specifically raised but also many others that we have considered in order to deliver what we think is a much improved version of our work. This version includes more paragraphs, English grammar revisions in all main sections, new references. Thanks a lot and happy new year. We are looking forward to your comments.

Sincerely,

Francisco-Javier Gil Mur

  1. The article is based on the study of the effect of titanium surface modifications on osteoblasts, ALP alkaline phosphatase activity as well as osteoblastic gene expression.

Overall, the work is interesting and quite well-written. It lacks information on the novelty of the research described therein. Moreover, it seems to me that the work should show not only single samples of titanium machined, sand-blasted and acid etching, but also the influence of the conditions of these modifications on the surface of titanium.

The results are well described, and the discussion is interesting.

The authors have added a paragraph in the introduction explaining the originality and objectives of this manuscript.

  1. There should be a Y-axis in the charts so that you can better track the values.

The Y-axis have been improved.

  1. The authors need to improve the way of representing decimal numbers - in English, decimals are written with a period, not a comma.

Done

  1. Authors should also standardize the bibliography - sometimes the names of journals are given in abbreviation, sometimes as full names.

The authors have revised the bibliography and have changed the title of the journal by their abbreviations according to the comment of the reviewer

  1. In general, I believe that authors wishing to publish a paper in the International Journal of Molecular Sciences should take into account the comments presented in this review.

Thank you very much for your comments that improve the manuscript. We have carried out all the reviewer's suggestions and comments.

Reviewer 3 Report

Dear Authors,

Presented work is written correctly, however Authors should introduce some corrections and improve some paragraphs especially concerning abstract, materials and methods results. These sections should be clearly written so that the reader can easily understand how the research was conducted and what for.

Abstract;

This part must be improved. Authors wrote “The roughness is higher in the GB and GB+AE samples with Ra around 1,2 μm compared to the AE (0,3 μm) and MA (0.02 μm) samples.”

What does the abbreviation “GB” mean?  Please explain.

Authors wrote: “The rougher samples have a more hydrophobic character: 750 for SB and 820C SB+AE in water compared to the AE (650) and MA (630) samples and have a higher polar component in the surface energy values (7,79 for SB and 8,18 mJ/m2 for SB+AE) compared to the AE samples (6,69 mJ/m2)”.

This sentence contains errors.

If the contact angle is clearly greater than 0°, but less than 90°, the surface

is hydrophilic, i.e. well wettable

What does it mean 820C?

What does the abbreviation “MA” mean?  Please explain.

Please explain also  abb. OPG, Runx2, RANKL.

Explanation is included in the introduction section however, it should be where the abbreviation was first used.

Keywords

In my opinion Authors did not investigate dental implants and their osseointegration with bone, only titanium grade 3 which might be used for dental implants.  Please replace these two by roughness and wettability and cells viability.

Introduction:

At the end of introduction, before the last paragraph: Authors should also characterize titanium as material for dental implants and write a paragraph about problems concerning dental implants justifying why surface modification of titanium implants is important.

Materials and Methods

Authors wrote twice:“The wettability of the four surface types was determined using the contact angle technique following the sessile drop technique under static conditions and at a temperature of 25°C.”

How many measurements have been done? If 3,  results are unreliable and Author should repeat measurements.

Also, Authors should do real statistics because SD for 3 measurements is not sufficient especially when the differences are not significant.

Results

Authors wrote” X-ray energy dispersive microanalyses showed the presence of aluminum and oxy-gen in the SB samples due to the use of Al2O3 as abrasive particle for the sand blasting treatment. These particles are embedded on the surface of the dental implant. This also occurs in the SB+AE samples although the amount of alumina decreases slightly due to the partial dissolution of the particles.” Where are these results? Where is methodology? Show BSE images and EDS analysis to improve it.

Fig 1 is not legible , especially for 3D profiles, scale bars should be visible.

Authors should also add comments of surface porosity and surface area which influenced on cells and bacteria adherence.

Fig 2, Authors wrote: “higher viability can be observed for the three days in the titanium discs with surface treated with alumina shot blasting and acid etching”. In my opinion no changes were observed.

Where is cells viability for control samples?

In what unit Authors measured cell viability? Should be added on the axis.

Fig 3 Authors included “morphology of cells” however, images should be taken in higher magnification because in my opinion images included in fig 3 only indicate numbers of cells on the titanium surface but nothing about flattening of cells. There is also nothing about samples preparation and cells fixation.

Where is control sample? Please comment SEM images.

Fig 4 Again, where is control sample?

Discussion and conclusion are fine.

Others errors:

All Latin words like “in vitro”, “in vivo” should be written in italics

Best regards and I hope I would be able to read your paper after improvement.

Author Response

REVIEWER 3

Dear Reviewer,

Thanks for taking the time to review our manuscript and suggest to us to improve our work by providing a lot more detail. We have done so, and we are now submitting a manuscript that not only addresses the points you specifically raised but also many others that we have considered in order to deliver what we think is a much improved version of our work. This version includes more paragraphs, English grammar revisions in all main sections, new references. Thanks a lot and happy new year. We are looking forward to your comments.

Sincerely,

Francisco-Javier Gil Mur

Dear Authors,

Presented work is written correctly, however Authors should introduce some corrections and improve some paragraphs especially concerning abstract, materials and methods results. These sections should be clearly written so that the reader can easily understand how the research was conducted and what for.

Abstract;

  1. This part must be improved. Authors wrote “The roughness is higher in the GB and GB+AE samples with Ra around 1,2 μm compared to the AE (0,3 μm) and MA (0.02 μm) samples.”

The sentence has been improved

  1. What does the abbreviation “GB” mean? Please explain.

This is a mistake. GB stands for Grit Blasting but in the paper we have written SB for Sand Blasting which is more common in the literature. We have changed GB to SB.

  1. Authors wrote: “The rougher samples have a more hydrophobic character: 750 for SB and 820C SB+AE in water compared to the AE (650) and MA (630) samples and have a higher polar component in the surface energy values (7,79 for SB and 8,18 mJ/m2 for SB+AE) compared to the AE samples (6,69 mJ/m2)”.

This sentence contains errors.

The sentence has been improved.

  1. If the contact angle is clearly greater than 0°, but less than 90°, the surface is hydrophilic, i.e. well wettable

The comment of the reviewer has been introduced in the text.

  1. What does it mean 820C?

It is a mistake. The sentence has been corrected. This is not a temperature value, it is a contact angle. Sorry.

  1. What does the abbreviation “MA” mean? Please explain.

Machined and its abbreviation is MA (introduced in the abstract). The authors have introduced an explanation.

  1. Please explain also OPG, Runx2, RANKL. Explanation is included in the introduction section however; it should be where the abbreviation was first used.

Done

Keywords

  1. In my opinion Authors did not investigate dental implants and their osseointegration with bone, only titanium grade 3 which might be used for dental implants. Please replace these two by roughness and wettability and cells viability.

Done

Introduction:

  1. At the end of introduction, before the last paragraph: Authors should also characterize titanium as material for dental implants and write a paragraph about problems concerning dental implants justifying why surface modification of titanium implants is important.

Done. A new paragraph in the introduction has been added according to the comment of the reviewer.

Materials and Methods

  1. Authors wrote twice: “The wettability of the four surface types was determined using the contact angle technique following the sessile drop technique under static conditions and at a temperature of 25°C.”

Corrected

  1. How many measurements have been done? If 3, results are unreliable and Author should repeat measurements.

Roughness was determined using ten samples for each surface studied. Three places for each sample were analysed (4 surfaces x 10 samples /surface x 3 places/sample). 120 measurements in total and 30 for each surface. This aspect has been clarified in the text.

  1. Also, Authors should do real statistics because SD for 3 measurements is not sufficient especially when the differences are not significant.

This issue has been commented in the previous question, since there are not three measurements but 10 samples for each surface that were analyzed in 3 zones of each of the samples. This fact was poorly explained in the text and has been corrected.

Results

  1. Authors wrote” X-ray energy dispersive microanalyses showed the presence of aluminum and oxy-gen in the SB samples due to the use of Al2O3 as abrasive particle for the sand blasting treatment. These particles are embedded on the surface of the dental implant. This also occurs in the SB+AE samples although the amount of alumina decreases slightly due to the partial dissolution of the particles.” Where are these results? Where is methodology? Show BSE images and EDS analysis to improve it.

This text has been added to the discussion with a more detailed explanation. These results refer to results obtained in another published scientific work in which the bibliographic citation has been added. These results are not part of this research work and this fact is made clear in the text of the discussion. We believe it is important to mention the effect of alumina on the surface of sand blasting treated materials.

  1. Fig 1 is not legible, especially for 3D profiles, scale bars should be visible.

The Figure has been improved.

  1. Authors should also add comments of surface porosity and surface area which influenced on cells and bacteria adherence.

A new paragraph has been introduced in the discussion about this aspect.

  1. Fig 2, Authors wrote: “higher viability can be observed for the three days in the titanium discs with surface treated with alumina shot blasting and acid etching”. In my opinion no changes were observed.

The reviewer is right in his comment and the authors have changed the text.

  1. Where is cells viability for control samples?

The control used are the as-received titanium discs corresponding to the MA as these have not undergone any treatment. This methodology is very frequent in biological studies of different titanium topographies. It has been explained in the text.

  1. In what unit Authors measured cell viability? Should be added on the axis.

Cell viability unit has been introduced in Y-axe and a explanation has been added in the text.

  1. Fig 3 Authors included “morphology of cells” however, images should be taken in higher magnification because in my opinion images included in fig 3 only indicate numbers of cells on the titanium surface but nothing about flattening of cells. There is also nothing about samples preparation and cells fixation. Where is control sample? Please comment SEM images.

The authors has been added a new Figure as an example and an explanation. In this Figure can observe the different morphologies of the osteoblasts.

  1. Fig 4 Again, where is control sample?

The control used are the as-received titanium discs corresponding to the MA as these have not undergone any treatment. This methodology is very frequent in biological studies of different titanium topographies. It has been explained in the text.

Discussion and conclusion are fine.

Other errors:

  1. All Latin words like “in vitro”, “in vivo” should be written in italics.

In vitro and in vivo have been written in italics.

Best regards and I hope I would be able to read your paper after improvement.

Thank you very much for your comments and for your time. Thanks a lot

Round 2

Reviewer 2 Report

Thank you for entrusting me with a manuscript review. The authors referred to my comments, except for one, the most important one.

In the first paragraph of my first review, I wrote:

Moreover, it seems to me that the work should show not only single samples of titanium machined, sand-blasted and acid etching, but also the influence of the conditions of these modifications on the surface of titanium.

This remark was omitted by the authors. I consider it very important and I ask you to respond to it.

Author Response

First of all, I would like to apologize to the reviewer for forgetting to correct this important point. I thank him again for his dedication and time to improve the quality of the manuscript and this point is very important.
The authors have added 4 paragraphs to comment on these variables and how they influence the behavior of dental implants and we have referred to the appropriate references to each of them. I hope, this new text is adequate for the reviewer. If you would like another approach or other aspects, we will prepare it according to your suggestions. 

Reviewer 3 Report

Dear Authors,

Your work looks much better now, but still you can improve it.

Abstract:

In the following sentence:

 The roughness values of the MA discs was 0.02 μm which increases to 0.3 μm with acid attack and becomes maximum for the sand blasted samples reaching values of 1.2 μm for Sb and SB+AE.

Please change Sb for SB

Introduction:

In the following sentence:

It is important to be able to obtain a titanium surface that once eliminated the bacterial bacterium can generate bone again.

Please change for It is important to be able to obtain a titanium surface which eliminated bacteria growing and stimulate bone regeneration

Materials and Methods

For WCA measurements there is still luck of information (How many measurements have you done?) Please add it.

Discussion:

In the following sentence:

This work is intended to help understand the influence of surfaces on osteoblastic behavior, but further research is needed to understand the biology of bone formation in implants.

Please change for This work is intended to help understand the influence of surfaces on osteoblastic behavior, but further research is needed to understand the biology of bone formation and its integration with implants

Good luck!

Author Response

Thank you very much again. The four comments have been coorected according to the reviewer.

Thank you very much for your help.

Round 3

Reviewer 2 Report

The Authors have corrected the text according to my comment.